# International Dynamic Marketing Capabilities of Emerging-Market Small Business on E-Commerce

Kihyon Kim  and Gyoogun Lim *

Department of Business Administration, Hanyang University, 222 Wangsimni-ro, Seongdong-gu, Seoul 04763, Korea; kihyon@hanyang.ac.kr
* Correspondence: gglim@hanyang.ac.kr

**Abstract:** For better export marketing strategies (EMS), companies mobilize their internal resources, which are managerial commitment, firm experience, and product uniqueness. However, Small businesses with constrained resources cannot be well explained with this view. So, more research on how small businesses come up with EMS has been called for. To explain how resource-restricted firms rely heavily on entrepreneurs, this study adopted the concept of dynamic managerial capabilities (DMCs) and resource versatility to better explain small business exports. We analyzed small businesses in Mongolia with qualitative research methods, including interviews with entrepreneurs and support organizations, site visits, and group discussions. We suggest international dynamic marketing capabilities (IDMCs), which are entrepreneurial orientation, networking capability, and versatile dynamic capability for small businesses. Theoretical and managerial implications are discussed.

**Keywords:** Cross-Border Electronic Commerce (CBEC); Export Marketing Strategy (EMS); International Dynamic Marketing Capability (IDMC); Dynamic Managerial Capability (DMC); entrepreneurial orientation; networking capability; versatile dynamic capability

## 1. Introduction

Cross-Border Electronic Commerce (CBEC; e-commerce), which is an online channel through which products are directly sold to international consumers via the internet, is expected to account for 22% of the total business-to-consumer (B2C) e-commerce worldwide by 2022 [1]. CBEC has profound impacts on a country's trade growth [2] especially emerging merkets [3]. It is a way to diversify against unstable situations in domestic markets in emerging markets for small businesses [4]. Therefore, for small businesses in emerging economies, CBEC is necessary for their export performance.

For a high level of export performance, export marketing strategies (EMS) are needed [5]. Researchers on factors that determine the development of EMS, internal resources of firms have been focused as main contributors [6]. Firms' internal resources, such as managerial commitment, firm experience, and product uniqueness lead the firm to develop EMS enhancing export performance [7,8].

However, small businesses have limited resources compared to large companies [9–11] and hindered in internationalization [5,12]. So, resource-based views cannot provide effective solution for small firmts to deveop EMS. Despite the increasing role of small firms in the world economy, export marketing research has focused too much on large firms [13] and research for small firms is still embryonic [5,14–19]. Also, only few studies have examined emerging market business [17,20]. There has been very little research on how small business entrepreneurs overcome resource deficiencies.

The aim of this paper is to explain how small firms in emerging markets adopt and develop EMS for CBEC with constrained resources. To fill the research gap and better strategize how small business can cope with EMS, we employed recent findings from extant research on small business entrepreneurs. Emerging-market small firms can overcome their

constrained resources by leveraging other capabilities [5]. For example, they can rely on entrepreneurs' capabilities [21,22], mobilize external partners' resources and apply existing resources in multiple functions. So, this study adopts the concept of dynamic managerial capabilities (DMCs), which contains variations of entrepreneurs' capabilities, including cognitive capabilities and networking capabilities [23–25]. In addition to this, we adopt the concept of resource versatility, which explains multiple uses of existing resources [26] to strategize how small firms adopt to challenges.

This paper adops qualitative research methods to analyze actual small business CBEC cases. We chose Mongolia, since there is a growing need in CBEC for small business to overcome its domestic market constraints, and relevant organizations are trying to support their EMS. With interviews and discussions with small business entrepreneurs and suppoorting organizations in Mongolia, this study presents better explanations on how small business in emerging markets adopt EMS and suggests practical strategies which can be replicated. It is timely since due to the current global pandemic, small firms need to strengthen their CBEC.This study also contributes theoretically in terms of filling the gap of EMS literatures which have been mainly focused on large firms.

## 2. Literature Review

### 2.1. International Marketing Capability

Firms mobilize their internal resources to develop EMS [1,7,8] and achieve competitive advantage [27,28]. Their internal resources, such as a firm's management commitment, firm experience, and product uniqueness have positive impacts on EMS [7,8]. To be more specific, highly committed managers are more likely to proactively adopt EMS [29,30] and more experienced firms are better at identifying market demands and understanding foreign markets [7]. Also, product uniqueness—the degree to which a product incorporates features to satisfy consumers' unique needs [31]—helps firms gain a competitive advantage in foreign markets [32]. (See Figure 1 for the conceptual framework of EMS.)

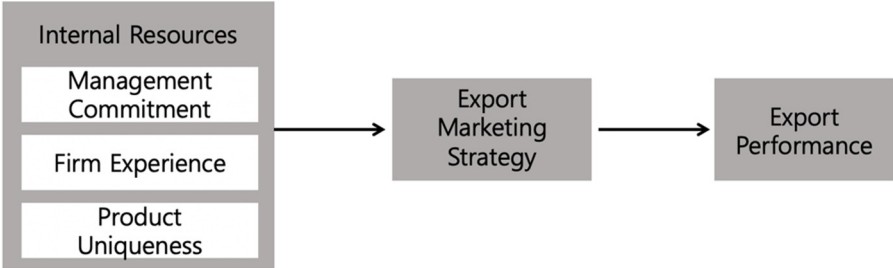

**Figure 1.** Internal determinants of EMS (partiallly excerpted from [8]).

However, small firms lack internal resources to develop EMS. For example, they are not convinced of the value of e-commerce [33] so lacks management commitment. Also, they have a low level of marketing experience [34,35] so lacks firm experience. Thus, they are at disadvantages to come up wtith EMS compared to larger entities [12]. It makes harder for them to expand beyond their domestic market [34,35] and be euipped for internationalization [36].

### 2.2. Need of Dynamism in Marketing Capabilities

To overcome these constraints upon them, small business takes various alternative measures [37–39]. However, internal resource-based views are static and do not apply dynamism in reality [22,40]. In real business case, dynamic capabilities, not just fixed resources positively influence CBEC [5,41] and firm performance [22,42]. In this regard, a firms' capabilities are not just the sum of internal resources [22,26,43]. A firm's capabilities are bundles of skills and knowledge that the firm has, based on diverse use of their resources [44].

Therefore, we applied the concept of International Dynamic Marketing Capability (IDMC), which well explains how a firm purposefully integrates, builds, and modifies internal and external resources [22,45,46]. For example, export-oriented capabilities include marketing capability [47], networking capability [48], and internationalization capability [49].

So, small businesses make use of dynamic marketing capabilities. Since most small firms are managed by a single owner [50,51], they rely heavily on the entrepreneur's capabilities [21,22]. Research on dynamic managerial capabilities (DMCs) well explain various entrepreneurs' capabilities [23,24]. DMC provides a useful theoretical lens to how various capabilities are converted to small business' resource. Small firms also need to incorporate external partners' capabilities to extend their resource bases [42,46,52]. DMC entails collaboration with external partners as well [23,25]. Because of its dynamism, DMC has gained popularity as a framework in recent years [53].

Small firms are more cross-functional than large firms. They modify certain capabilities and applied them in multiple roles. This means that existing constrained resources are more versatile than their larger counterparts [25]. Research on resource versatility, which well explain this dynamism, can be applied to small busienss. So, this study incorporates extant research on DMC and resource versatility. More detailed explanations are following.

### 2.3. Entrepreneurs' Dynamic Capability

Managerial capability is how managers build, integrate, and reconfigure their organizational resources [54]. It includes managerial cognition; personal beliefs and mental models for decision-making [54,55]. This affects how the firm senses market changes and behave subsequently. The entrepreneurs' capabilities to proactively seeking new markets lead to better EMS and export performance [56], whie inert managerial cognition fail to recognize market changes resulting in poor performance [55]. Entrepreneurial orientation refers to the capability of the entrepreneur to be proactive in seeking new markets [3,57] and it leads to development of better EMS [5,58].

Hence, we propose the following:

**Proposition 1.** *For emerging market small business, entrepreneurial orientation leads to the development of EMS.*

### 2.4. Networking Capability

Network refers interconnected firms and managers involved in economic activities that convert resources into outputs [59,60]. It encompasses external partners [61], such as technological alliance partners [19,38,62] and even informal relationships [63]. Firms leverage networks to renew their resources, develop marketing capabilities and strategies [5,32,64], obtain competitive advantages to achieve business goals [65]. In exporting, local partners perform even core activities such as marketing [5] and identifying new market opportunities [66]. Especially in the early stage of internationalization, leveraging external partners' capabilities is essential [38,67,68]

For small business, entrepreneur's network capability determines the firm's network [69] and entrepreneur's network helps the firm to obtain resources [56], compensates scarce resources [70] and reinforces capabilities [71]. In terms of EMS, entrepreneurs' formal and informal networking activities function as marketing activities [19,56,64] and help the firm to develop better marketing strategies [5].

Hence, we propose the following:

**Proposition 2.** *For emerging market small business, networking capability of entrepreneurs leads to the development of EMS.*

### 2.5. Versatile Dynamic Capability

To overcome scarcity of resources, small businesses use existing resources in more effective and efficient way. They modify certain capabilities to apply them for multiple roles [9].

For example, marketing capabilities can be used in product development since capabilities overlap [39]. To explain how small business can mobilize their limited resource for better export performance, we adopt the concept of resource versatility, which conceptualizes how certain resources support multiple roles in resource-constrained firms [54].

Hence, we claim the following:

**Proposition 3.** *For emerging market small business, versatile dynamic capabilities have a positive impact on the development of EMS.*

To summarize, Figure 2 shows how emerging market small firms can be successful in foreign markets using their capabilities, which termed as international dynamic marketing capabilities [6,45]. Adopting research on managerial dynamic capability theory and resource versatility, we propose that entrepreneurial orientation, networking capability, and versatile dynamic capability might help small business to develop better EMS for better export performance.

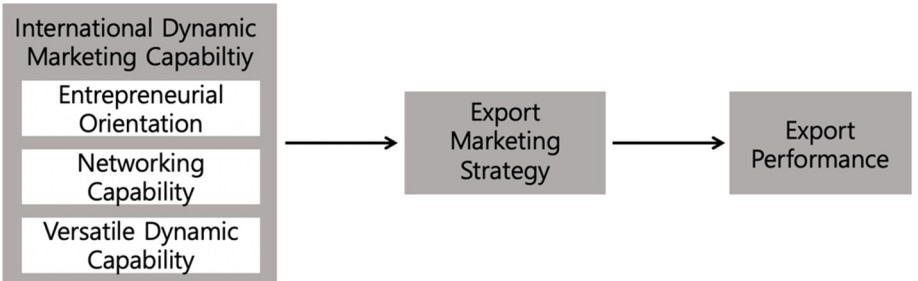

**Figure 2.** International dynamic marketing capabilities of emerging-market small business.

### 3. Research Methods

*3.1. Research Design*

The aim of this study is to explore how small business in emerging economy mobilize their capabilities to develop EMS in CBEC context. We adopted qualitative study methods, because they can be used to derive findings from under-studied phenomena [72] and to understand processes that could not be disassociated from context [73]. This is also to compare new empirical findings with those of prior studies [74]. In this sudy, how international dynamic maketing capabilities of small business are used to come up with better EMS is compared to internal resources of large firms [28,75,76].

*3.2. Data Collection*

For small business case analysis, we selected micro-enterprises in Mongolia for several reasons. First, Mongolia is an emerging economy with growing e-commerce. Not just its domestic e-commerce industry shows unprecedented growth; its revenue increased more than 60% during 2020 [77]., it also has a relatively good infrastructure for e-commerce in the capital city, Ulaanbaatar, as emerging economy. Of the country's total population of 3 million, 2.6 million have internet access, while 2 million use smartphones, and 1.3 million use social networking services (SNSs) on a regular basis [77]. All these numbers show Mongolia's potential for CBEC. Also, CBEC is necessary for Mongolia to overcome the challenging geography of the country for trade groth. Thus, there are demands from small business to improve their export. So, international non-governmental organizations (INGOs), foreign and domestic governments are taking initiatives to support small business entrepreneurs doing CBEC. One of the examples is the business incubation center (the Women's Business Center) in Ulaanbaatar. In 2016, it was established by the support from the INGO (the Asia Foundation), the Korea government (Korea International Cooperation Agency), and the Mongolia government. They work together to support female micro-entrepreneurs and to incubate successful small business.

We collected data from multiple sources for triangulation [72]. We could implement internal documents reviews, interviews, site visits, and focus-group discussions with entrepreneurs, staffs of the business incubation center and many other relevant parties in Mongolia, Korea, and the US. To select representative small businesses, the following criteria were applied: the business (a) located inside the Women Business Center; (b) with less than 15 employees; (c) have experience of domestic e-commerce; (d) currently involved in CBEC. We selected three representative cases with the help of the Women Business Center.

To enhance the feasibility and practicability of the research, we interviewed colleagues from external partners of small businesses; the Asia Foundation Mongolia office which runs the Women Business Center; IT start-up incubation center in Ulaanbaatar which partners with the center; digital finance service firm which work closely with the center; and the bank in Ulaanbaatar. To ensure the research findings, we also interviewed people who have experience in exporting Mongolian goods to Korea. Table 1 shows the details of the interviewees.

**Table 1.** Information about the Interviews.

| Case | Interviewee | Nationality | Business | Interview Occasion |
|------|-------------|-------------|----------|--------------------|
| A | Entrepreneur | Mongolia | Eco-friendly flowerpots | 9 September 2019/Ulaanbaatar |
| B | Entrepreneur | Mongolia | Organic foods and beverages | 9 September 2019/Ulaanbaatar |
| C | Entrepreneur | Mongolia | Hand-crafted accessaries | 9 September 2019/Ulaanbaatar |
| D | Entrepreneur | South Korea | IT start-up incubation | 16 August 2019/Seoul |
| E | Manager | Mongolia | ITstart-up incubation | 10 September 2019/Ulaanbaatar |
| F | Head of Corporate Sales | Mongolia | Digital finance service | 10 September 2019/Ulaanbaatar |
| G | Former CEO | Mongolia | Banking service | 10 September 2019/Ulaanbaatar |
| H | Country Representative | US | INGO office in Seoul | 18 July 2019/Seoul |
| I | Manager | US | Business incubation center | 27 August 2019/Seoul |
| J | Manager | Mongolia | Business incubation center | 9 September 2019/Ulaanbaatar |
| K | Manager | Mongolia | Business incubation center | 9 September 2019/Ulaanbaatar |
| L | Former Manager | South Korea | Korea government | 26 June 2020/Seoul |
| M | Former Manager | US | INGO office in Ulaanbaatar | 20 September 2019/Washington D.C. |

At each interview, people were encouraged to speak freely with no forced answers. The interviews were conducted in Mongolian, Korean, and English and the average length was about one hour. With entrepreneurs, we tried to understand how small firms develop EMS and tried to confirm it with external partners. After the completion of the research, we contacted external partners again to check their views on feasibility and applicability of research findings based on their experience. Several findings were actually adopted by the center to support small business doing CEBC.

*3.3. Data Analysis*

For data analysis, two researchers repeatedly reviewed the interview results and reach a consensus [78]. Then the results were grouped into categories. We shifted back and forth between the interview results and the broad scope of literature to reach a consensus, which is supported by theories. This analysis was similar to Strauss and Corbin's notion of open coding [79]. For example, interview results on small business' challenge were categorized according to theories. Interview results on possible solutions were also categorized based on theories. Then those were integrated into a theoretical concept. The consensus and findings were reported back to the center to ensure validity and practicality. The level of disagreement was generally low. Through these processes, we tried to match the interview data with theories [79]. Figure 3 explains the data structure on interviews and literature reviews.

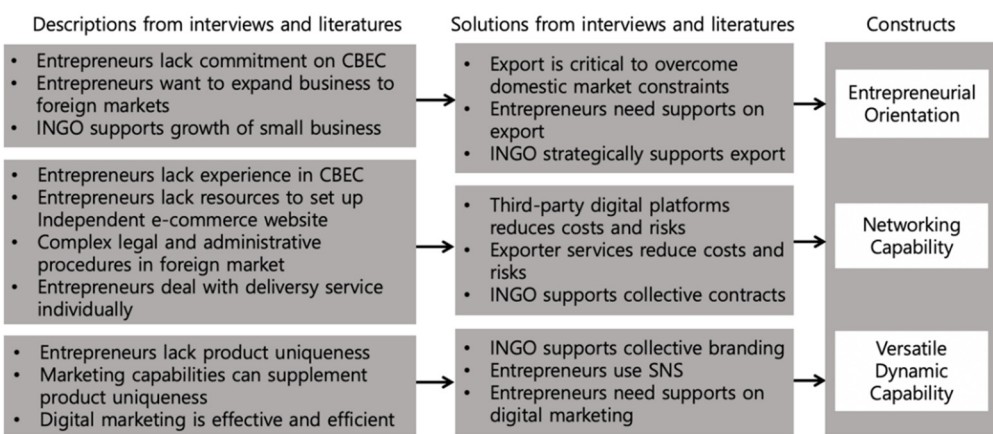

**Figure 3.** Data structure of interviews results and literatures.

## 4. Case Findings

### 4.1. Managerial Commitment and Entrepreneurial Orientation

High level of managerial commitment is one of the internal resources that leads to the development of better EMS [29,30]. However, small firms are not convinced of the value of e-commerce [33] so lacks management commitment. According to the interviews, most of the entrepreneurs were not committed to CBEC. For them, although they had experience in domestic e-commerce, cross-border presents unprecedented challenges that they are not acusstomed to. These entails export procedures, such as customs and cross-border deliveries. However, entrepreneurial orientation, which is the capability of the entrepreneur to be proactive in seeking new markets [3,58], can be facilitated for the development of EMS [5,59]. According to the interviews, entrepreneurs want to expand their business and seek new business opportunities. Despite a low level of commitment to export, small business entrepreneurs have entrepreneurial orientations to expand their markets to foreign countries:

*"I want to export the product abroad. [ . . . ] However, I never experienced selling products abroad [ . . . ]. It is difficult for me to know about export procedures [ . . . ] It is hard to get necessary information regarding export [ . . . ] It is challenging for us [ . . . ] I hope that I can get support."* (Entrepreneur, Respondent A)

*"We are currently selling organic juice essence to Japan [ . . . ] We also want to expand to [the] Korea market. But we need to consider that export entail taxes, deliveries and other complex procedures [ . . . ] It is still challenging for us."* (Entrepreneur, Respondent B)

*"We want to sell hand-crafted goods to foreign markets [ . . . ] We need to check whether it is possible or not [ . . . ] We need consultations on this [ . . . ]"* (Entrepreneur, Respondent C)

So, we confirm that entrepreneurial orientation, rather than managerial commitment, needs to be considered as a factor influencing the devlopment of EMS. This was confirmed by interviews with external partners including managers of the business incubation center, former CEO of the bank, head of the digital finance service provider, and former government officer. They confirmed that CBEC is critical for the growth of a small business. Managers of the business incubation center explained how they share other entrepreneurs' successful CBEC and facilitate entrepreneurial orientation:

*"We strategize that micro-entrepreneurs from developing countries need to sell their products abroad [ . . . ] since their local market size has limitations for growth [ . . . ] export through e-commerce opens windows of opportunities for them [ . . . ] Also, trades are now moving to e-commerce [ . . . ] So we are putting our efforts to effectively to support export using e-commerce."* (Country Representative of INGO, Respondent H)

*"We plan to build up e-commerce websites which sell small business products [ . . . ] We can offer digital platforms for micro-entrepreneurs to sell their products [ . . . ] Not just production—we want to support selling products."* (Manager of the business incubation center, Respondent I)

*"We run programs for entrepreneurs based on their needs . . . such as sharing the experience of other successful exporters [ . . . ] Once we invited entrepreneurs from Korea to share their experiences in exporting [ . . . ] Through those experiences, entrepreneurs were motivated to export."* (Manager of the business incubation center, Respondent J)

### 4.2. Firm Experience and Networking Capability

Small businesses have a low level of marketing experience [34,35]. Small firms we interviewed expressed their concerns about their lack of knowledge and experience:

*"In the domestic market, most of our products are sold online [ . . . ] However, I concerned that I never experienced selling products abroad."* (Entrepreneur, Respondent A)

One of the main challenges that small businesses face is how they adapt to CBEC platforms. Large companies usually develop their own digital platforms [80,81]. It takes a lot of resources to set up their own digital platforms. So, small firms utilize existing platforms instead of making investments to set up their own platform [82]. Third-party digital platforms provide easy access to international markets [24] and help take advantage of existing industrial chains, trade processes, payment methods, logistic, and warehousing. They also reduce risks and transaction costs, because of their trust [83]. So, it is recommended for small firms to rely on third-party digital platforms such as Amazon and Alibaba [84]. They provide small exporters with technological tools for analysis [85], payments, and even logistics services [86].

To validate these recommendations in the Mongolia context, we analyzed CBEC platform options for Mongolian entrepreneurs to export to the Korean market. Setting up their own independent platforms required legal and administrative procedures, including business registration and e-commerce registration in Korea. However, third-party platforms, such as "G-market Global Shop" (Korean e-*commerce* platform), do not require this procedures. This open-market platform aims to provide services to less-resourced independent sellers. IT service providers in Mongolia agreed and the business incubation center decided to use existing digital platforms, rather than setting up their own:

*"I recommend using Korea open markets [ . . . ] We do not recommend making an independent website [ . . . ] Because micro-enterpreneurs do not have the expertise, experience, and enough resources to set up and manage a digital platform [ . . . ] They may face difficulties in managing it."* (Entrepreneur, Respondent D)

*"So far, we planned to set up independent websites for micro-entrepreneurs [ . . . ] However, it seems that we should rather focus on supporting entrepreneurs entering into existing market platforms [ . . . ] It will save lots of resources [ . . . ] It will be more practical and effective [ . . . ]"* (Manager of the business incubation center, Respondent I)

Another chlallengee that small firms face is exporter service. Exporter service includes deliveries, customs, and administrative procedures to export. Currently interviewed entrepreneurs individually deal with exporter services. However, these procedures can be outsourced to specialized agencies [83]. Outsourcing improves service quality, reduces costs, and enables entrepreneurs to focus on their core business [87]. So, after interviews and discussions, we suggested small businesses to make a collective contract to use exporter services, since individual contract may be challenging for them because of its small size. For example, multiple small firms can make collective contracts on distribution:

*"It is plausible to use delivery services collectively [ . . . ] There is an existing logistic service exporting Mongolian products to Korea run by exporter company [ . . . ] Usually,*

*they make a contract with big exporting companies [ … ] Likewise, Mongolian entrepreneurs can cooperate with each other and make collective contacts."* (Entrepreneur, Respondent D)

Government and INGOs can support collective contract between exporters and multiple small firms. This strategy is implemented in more advanced countries: Korea government provides collective delivery services for Korean small businesses.

### 4.3. Product Uniqueness and Versatile Dynamic Capability

For small business, satisfying the unique needs of customers in foreign markets is challenging. Without differentiated products with competitiveness, it is hard to persuade foreign consumers to buy the product. According to the interviews, Mongolian products are also challenged by product uniqueness. One of interviewee who has exported Mongolian products to Korea shared experience. Only a few product categories have product uniqueness:

*"Mongolian manufactured products are competing with Chinese products in foreign markets [ … ] In many cases, it is challenging to have a competitive advantage [ … ] Rather than manufactured products, there are several categories that have competitiveness in foreign markets [ … ] Several products made from natural ingredients have high quality and good perception in Korea [ … ] For example, natural ingredients such as honey [ … ] cashmere have a competitive edge [ … ] However, they have other challenges such as food standards [ … ] producer associations who deter imports in foreign countries [ … ] In Korea, selling Mongolian honey was tried several times but faced challenges."* (Entrepreneur, Respondent D)

*"Products which are exported to nearby markets, including Korea, China, and Japan, are cashmere, wool, meat, and salt [ … ] They have competitiveness."* (Manager of the business incubation center, Respondent I)

According to the resource versatility, product uniqueness can be supplemented with marketing capabilities. Innovative marketing enables small firms to differentiate their products [69]. For example, branding, which entails names, signs, symbols, or designs, differentiates the products in export markets [87]. Therefore, we suggest small businesses to collectively develop branding strategies. For example, collective branding such as the "Mongolian Natural Cashmere Initiative" can be developed. Such strategies are already implemented in similar cases by the United Nations Development Programme (UNDP) for the purpose of supporting local products. In this case, the supporting organization works as a guaranteer of product quality:

*"I think collective branding can be a good solution for entrepreneurs [ … ] In cases of social enterprises in the US or Europe, they are supported by certifications or collective branding from external organizations [ … ] Consumers are convinced of the quality of products."* (Former manager of INGO, Respondent M)

*"Collective effort would be more effective and efficient [ … ] Contracting collectively with Korean marketing agencies is required, rather than contracting individually."* (Manager of government, Respondent L)

Small business can employ innovative communication with small budgets, [51,88]. They can utilze less-resource-intensive open-source solutions [89], such as SNS and search engine advertisements [41]. Digital marketing leads to better export performance [1,51], since it helps to build long-lasting customer relationships [17,90]. For example, social media enables word-of-mouth communication and increases sales [91]. According to interviews, entrepreneurs actively use SNS as a digital marketing tool. Interviewees expressed their needs of support on digital marketing:

*"We get orders and feedbacks using Facebook [ … ] It is the most convenient way of doing e-commerce with little expense."* (Entrepreneur, Respondent A)

*"Even specific skills such as PhotoShop and SNS marketing would be helpful."* (Entrepreneur, Respondent C)

*"Workshops on how to implement online marketing would be helpful."* (Entrepreneur, Respondent B)

*"Entrepreneurs like practical workshops regarding SNS [ ... ] For example, simple skills such as taking photos and uploading them to an SNS with a basic design help their e-commerce business."* (Manager of the business incubation center, Respondent K)

## 5. Discussion

Based on the resource-based view, internal determinants of EMS are management commitment, firm experience, and product uniqueness [8]. This cannot be extended to small businesses, they lack resources. Also, this perspective is static [22,40] and does not contain dynamic capabilities that affect export [5,22,41,42]. So, this resource-based view does not apply to small firms from emerging economies that are doing a critical role in the economies of developing countries.This study adopted findings from extant research on dynamic managerial capability and resource versatility.

By merging findings from multiple data sources, including interviews with entrepreneurs and support organizations, this paper confirmed research findings regarding new components of determining factors of EMS for small firms. It replaces the existing internal determinants of EMS with new components: entrepreneurial orientation, networking capability, and versatile dynamic capabilities. In terms of entrepreneurial orientation, Mongolian entrepreneurs are motivated to enter the Korean market. Networking capability enables Mongolian entrepreneurs to use third-party marketplaces rather than develop their own independent websites and use exporter services. Also, they can adopt collective branding strategies and use digital marketing tools to complement their products' uniqueness. Those factors have proven to be more adequate to explain the development of EMS for small companies.

## 6. Conclusions

Theoretically, this study presents a fresh point of view on how small business develop export marketing strategies. It presents a procedural approach with dynamism rather than a static approach based on a resource-based view. Scholars have called for research on procedural view based on effectuation theory [53] and dynamic marketing capability [73]. Whereas literature often treated "capability" as static, our study shows how capabilities can be used in dynamic and multiple ways. This versatility approach well explains how flexibility and fast learning of small business can be leveraged in CBEC. This is meaningful in terms of responding to those requests.

This research is also valuable in terms of managerial implications. If governments or INGOs seek effective ways to enhance exports of small business, they should not approach based on resource-based views. Unfortunately, current policy prescriptions often focus on the creation of property-based resources [92]. Instead, they need to facilitate small businesses connecting with foreign partners [93]. Small firms can work with e-commerce platforms, third-party logistics services, and local marketing agencies and learn from interactions with foreign customers.

There are limitations in this study. First, our examination was conducted in the unique context of Mongolia and Korea and data were collected only from the capital city of Mongolia, which may not represent the whole country's population. Secondly, we relied on interviews with a relatively small number of informants. Even though we tried to enhance the interview data with interviewees who have rich experience, we have a limited number of business cases. Third, we only focused on the capability perspective, which does not address other aspects, such as organizational structure and routines.

Future studies could be extended to a more diversified group of entrepreneurs, such as rural producers, other countries, and other markets, which would provide the possibility for comparison with the role of IDMC in small businesses operating in other markets. The

collection of additional case studies and survey data would enhance the generalizability of the results. Also, to confirm each determinant, quantitative research is needed.

**Author Contributions:** Conceptualization, K.K.; methodology, K.K.; validation, K.K. and G.L.; formal analysis, K.K.; investigation, K.K. and G.L.; resources, K.K. and G.L.; data curation, K.K. and G.L.; writing—original draft preparation, K.K.; writing—review and editing, K.K.; visualization, K.K.; supervision, G.L.; project administration, K.K.; funding acquisition, K.K. and G.L. All authors have read and agreed to the published version of the manuscript.

**Funding:** This research was partly supported by The Asia Foundation. This work was supported by 446 the research fund of Hanyang University (HY-201800000001046).

**Institutional Review Board Statement:** Not applicable.

**Informed Consent Statement:** Informed consent was obtained from all subjects involved in the study.

**Acknowledgments:** The authors would like to thank the anonymous reviewers for their insightful comments and remarks which enabled correcting the article.

**Conflicts of Interest:** The authors have no conflict of interest to declare.

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
