# Peer review of "International Dynamic Marketing Capabilities of Emerging-Market Small Business on E-Commerce"

_jtaer, doi:10.3390/jtaer17010010_

Round 1
Reviewer 1 Report
The paper, “Supporting Cross-Border E-commerce of Micro Entrepreneurs in
Developing Countries: Export Marketing Strategy”, addresses a research area interesting in marketing area. However, some aspects should be considered. For example, the Introduction section, the authors authors draw on some prior studies, but a much more critical literature analysis is needed to strengthen the paper’s argument and draw out the gaps they seek to address. Which gap(s) in extant studies are authors trying to address here? That means, the introduction should clearly indicate the need for this paper in relation to extant research studies.
While the author(s) establish some links to some literature, author(s) need to establish a more coherent framework for the overall paper and more recent literature can be used to support the research topics (create a new section titled Literature Review). The theories used could be reinforced to support the study’s development. Strong theorizing remains scarce. It is this strong theorizing that is needed to explain the phenomenon here proposed.
You used multiple case study but criteria selection about the three firms are necessary. A characterization about the interviewees is also necessary and why you select them. Also, the paper needs to be present much stronger discussion and conclusion sections in order to offer value to the reader. Overall, the manuscript makes some interesting points, but the study aims to study some relations among the variables. As such, we have a better chance of making a contribution to theory, which is sought. You do not situate your study in a theoretical frame.
While you have made a valiant attempt to tackle these topics, I would argue that the data used is “good”, but the authors should use more robust analysis to support the empirical study and to achieve the main research objective. However, the results obtained should be discussed with more comparative previous! Such a discovery should be informing you that there is little if any novelty in your empiricism. find?
The conclusions and implications could be extended, innovative and more contributions for managers and public policy should also be presented.!
Author Response
Thanks to reviewers’ comments, we could furture develop research with adoptino of current model of contingency model of exportmarketing strategy. Compared to previously submitted version, this revised paper has written almost with major revisions, with a new research framework. It leads to clear theoretical implications to fill research gap, which reviewers mentioned to clarify. Bacause the paper was rewritten totally, revisions following revewers’ comments are breifly described here. Detailed changes and revisions are found on the paper.
Reviewer 2 Report
Cross-border e-commerce is an important trend in the current retail industry, and there is indeed a need for further research. But after all, this is an academic paper. Based on academic requirements, I put forward the following review comments.
- This manuscript needs to be revised drastically in the literature. First, the authors lack a comparative analysis of cross-border e-commerce in developing countries in recent years. Secondly, the literature referred to by the authors lacks recent research results. Many references are articles from decades ago and cannot be used as references for current e-commerce development. Third, the theoretical basis adopts the 4P strategy. However, the 4P strategy is already a basic condition in marketing. The authors only use this theory as the basis for their analysis, and cannot be academically enlightening. I suggest that the authors re-arrange it in the literature.
- In the “Materials and Methods” section, the content from 2.1 to 2.7 should belong to the section of literature review.
- In line 119, please change "(Yudelson 1999)" to the reference code.
- In line 149, please change "(Yin, 2008)" to the reference code.
- In line 161-164, the authors stated "As of 2019, WBC has 3,700 female members and is incubating 47 entrepreneurs. In its first phase between 2016 and 2018, 39 entrepreneurs have completed WBC's incubation programs. The main businesses are hand rafts and service." The author is asked to provide the source of the literature.
- In Table 1-3, authors should explain and mark the role of Table 1-3 in the description of this article.
- In line 180 and 292, "1" is redundant.
- In Table 1 and Table 2, the representation of "Title 1, Title 2, Title 3" in the header of the table is inappropriate. Please correct it.
- In Table 2, please delete unnecessary horizontal lines.
- In the “Result” section, I cannot understand the relationship between the authors’ research results and data collection. Qualitative research has a certain way of writing. The authors are requested to cite the results of the data collection in due course.
- In line 226-227, the author explains "However, according to literature, it is recommeded to use third party platforms since there are positive impacts of using them." Please provide the source of the literature. In addition, the "recommeded" of this text is a misspelling.
- In the “Discussion” section, the authors did not compare with previous studies, so it is difficult to understand the value of this article. In addition, the authors should also have an “in-depth” discussion of the research results.
- There are too many spelling and grammatical errors in the authors' manuscripts. There are too many errors, so I can't point them out one by one. The authors are requested to carefully correct the manuscript.
- In the “References” section, there are too many errors in format, and even incomplete reference source writing (eg: [20] and [37]). There are too many errors, so I can't point them out one by one. The authors are requested to carefully correct the manuscript.

Author Response

(The authors gave the same response as above.)

Reviewer 3 Report
The subject of the article is interesting but some improvements are needed. Please find below my remarks:
- Introduction: The article does not have explicitly stated a goal or a research question. No objective or hypothesis is defined for this research. It makes the rest of paper with lack of coherence, mainly in the results presentation. Even if there is a subsection (2.6) dedicated to the research question, a veritable question and research objectives are not established.
- The subsections 2.1-2.5 are merely a literature review, which is not suitable for the methodology section. Maybe a more detailed description of the e-commerce sector in the analysed country should better fit in this section.
- In subsection 2.8, multiple sources of data have been mentioned but they cannot be fully compatible. Each source could be used for different researches that have different objectives and methodologies. OK, they could be used for Data triangulation but the main research should be mentioned and fully described. I understand that the main research was a semi-structured interview. But which is the relationship with the case studies mentioned in the subsection 2.7? There is no case study included in this paper.
- The size of the sample used for interview is not mentioned anywhere. From the Table 1 it results that 17 people were interviewed. The selection criteria were not mentioned. Also the Table 1 does not have the titles of the columns. The same situation for Table 2.
- At Row 212 focus group discussions are mentioned but it is not very clear who participated to this group and which are its objectives and results.
- The Results section is very poor explained. In the absence of research objectives the subsections presented at results are meaningless. It is not very clear if the results are based on the findings from interviews or on literature or on the authors’ experience. I consider that the results should be based on the information provided by the interviewees. It should be mentioned very clear that the results represent the opinions of these people.
- In Table 3 it is presented a very original SWOT analysis. As far as I know, this analysis is not a Crosstab.
- In the Discussion and Conclusion sections the contribution of the research should be emphasized. Maybe here should be detailed the EMS and sustained by the research findings and literature review.
- The implications of the research results for theory are not explained in the conclusion.
- At Row 366, the case study is mentioned again but this paper does not contain any case study. The fact that only one country was analysed does not mean that a research based on case study method was conducted.
- At Row 397 maybe a fragment from the Template was lost in the article. Future research directions are not enough described.
I hope that authors should make major revisions to the article in order to make it publishable in this journal.
Author Response

(The authors gave the same response as above.)

Reviewer 4 Report
While this article offers interesting results, some revisions are required:
- in the introduction, the authors should provide more information about the role of e-commerce and they should also develop the growing of e-commerce in the context of the pandemic.
- added to the challenges that MSMEs face, the introduction should also engage with previous research and what has been addressed before. Also, the authors should identify properly the research gap.
- the authors should better analyze the marketing mix and provide a better understanding of digital marketing for SMEs.
- the section 2.1 about export marketing strategy needs more background to support the claims. This also happens with the development of the 4P’s of marketing mix.
- in 2.7 the research framework is not enough developed, and the authors must provide the rationale of the method used.
- the authors analyse Mongolia MSMEs exporting products to Korea, and the reasons must be supported with references.
- similar weaknesses are observed in 2.8, where the authors should improve the description of the method, and explain the processes, in order to justify the selection of interviewees, how observation was conducted, or how focus group discussions were developed.
- results are too simplistic, the authors must develop a critical approach to the marketing mix strategies and add quotes from the interviews.
- please revise headings of columns in tables 1 and 2.
- discussion should provide dialogue between results and literature. The authors should revise the discussion or merge results and discussion.
- the authors should develop the last paragraph of the conclusion about opportunities for further research.
Author Response

(The authors gave the same response as above.)

Round 2
Reviewer 1 Report
The paper presents several weak points and flaws.
Author Response
Based on the comments of reviewers, the authors made substantial improvements of the research with new research framework. Based on recent export maketing researches, research gap and aim of the study has been corrected. Following reviewers’ comments, each section of the paper has been rewritten.

Reviewer 2 Report
The authors have made substantial amendments to the content of this manuscript. However, JTAER is still an academic journal with academic rigor. So I still review the authors' manuscripts from an academic perspective. 1. The entire manuscript does not look like a result of academic research, but rather like a work report. Authors are asked to write manuscripts in accordance with the basic requirements of academic journals. 2. In the “Introduction” section, the authors did not clearly write out the research question. Research problems are usually caused by insufficient research in the past. The authors did not clearly state this point. In addition, the reasons for the authors to adopt the contingency model are not clearly stated. This makes it impossible for readers to know what enlightenment they can get from reading this article. 3. In the "Literature Review" section, the authors' manuscripts have the following contents that need to be greatly improved: (1) The contingency model is a theory that was put forward decades ago, and there are still a lot of researches on this model in recent years. It is a pity that the authors did not make in-depth literature discussions on the contingency model, especially for comparative analysis of a large number of studies in recent years, so that readers cannot understand the applicability of the contingency model in this research. In addition, the author uses this model as the theoretical basis, and does not clearly explain why this theory is used. In fact, there are many theories to support the export marketing strategy, and perhaps the contingency model is not an appropriate theoretical basis. Authors can rethink. (2) Even though the authors think that the contingency model is a good analysis framework, the content of the entire "Literature Review" section is regarded as an "inevitable" practice in the marketing management field. I can't understand how this kind of content can inspire readers. (3) The entire "Literature Review" section is written in a columnar style, which prevents readers from understanding the author's thinking. The entire "Literature Review" section looks like a briefing material, not an article. Please correct it. 4. In the "Research Methods" section, the authors focused on MSME (Micro, small, and medium-sized enterprises), but the authors only visited 5 MSMEs. Obviously, these interviewees are insufficient, and the authors did not state the representativeness of the interviewee. Instead, the authors interviewed many public sector personnel, but the interviews with these public sector personnel did not effectively reflect the research content. This shows that the authors have flaws in the selection of interviewees. 5. In the “Results” section, the research results of this manuscript are intuitively obvious to readers. The authors believe that the biggest contribution is to revise the contingency model. But I don't understand why the author wants to correct it? How to fix it? What's wrong with the past model? Is the revised applicability more theoretically malleable? These should be the focus of this article. But I did not find it in this manuscript. 6. In “Discussion” section, this is the most important section of the entire manuscript. However, the authors did not make an in-depth discussion, nor did they make a comparative analysis of past research results. The authors are just restating the results. Such writing does not meet the academic requirements of the "Discussion" section. 7. Figure 1 and 2 do not appear in the text of the manuscript, nor are they discussed or explained. This makes readers unable to understand the purpose of these two Figures. 8. There are too many spelling and grammatical errors in the authors' manuscripts. There are nearly a hundred of these errors. Even the simplest spelling can make mistakes. This is beyond the scope of the reviewer's assistance. In addition, in the "References" section, the authors have omitted a lot of research in recent years, and there are also many errors in the format. Authors are requested to pay attention to the rigor of writing academic articles. 9. This journal is JTAER. Why do authors use the template of "Sustainability" journal?
Author Response

(The authors gave the same response as above.)

Reviewer 3 Report
The authors made significant improvements to their work but in my opinion some clarifications should still be performed.
- The aim of the paper or the research question should be clearly stated in the Introduction. It is not enough to say: "This paper tries to bridge this gap between current contingency model and MSMEs’ cases". Maybe some variables or strategies should be identified in order to bridge this gap.
- In the Conclusion section the implications for theory are still not included. Also for the managerial implications, the authors should better describe the strategies that SMEs can put in practice in order to develop their activities. It is not enough to say: "MSMEs can focus on investing resources on each factors of this study". This is a poor explanation of the managerial implications.
Author Response
Based on the comments of reviewers, the authors made substantial improvements of the research with new research framework. Based on recent export marketing researches, research gap and aim of the study has been corrected. Following reviewers’ comments, each section of the paper has been rewritten.

Reviewer 4 Report
While the authors have improved the paper, some revisions are required:
- in the introduction, the authors say ‘this paper tries to bridge this gap between current contingency model and MSMEs’ cases’ and they should explain how they plan to fill the gap.
- the methodology is very important for the quality of the paper and the authors should add points to accurately develop the processes and techniques (site visits, field observations, semi-structured interviews, focus groups and reviewing data).
- the authors made a remarkable effort in the revision of the results, however some quotes should be critically discussed.
- the discussion is not sufficiently improved and the authors should expand the dialogue between results and literature.
Author Response

(The authors gave the same response as above.)

Round 3
Reviewer 1 Report
The authors improved the paper, but some sections could be reinforced.
Author Response
Based on the comments of reviewers, the authors made substantial improvements of the research with new research framework.

Reviewer 2 Report
The authors have made substantial amendments to the content of this manuscript. However, JTAER is still an academic journal with academic rigor. So I still review the authors' manuscripts from an academic perspective. Especially when I reviewed the comments for the second time, the authors hardly made any improvements. So I repeat it again.
- The entire manuscript does not look like a result of academic research, but rather like a work report. In particular, academic papers first focus on theoretical exploration and enlightenment. The authors' manuscripts are obviously insufficient in this regard. For example: on line 39-41, the authors said: "this study employs recent findings". However, most of the literature and concepts are more than 10-20 years old, and it is necessary to add "new" theoretical views.
- In the "Literature Review" section, the authors still haven't clearly stated the problems of the previous "contingency model" at the theoretical level and the necessity of revising the "contingency model". Please elaborate. Although the authors stated that they have abandoned the contingency model view, the description of the contingency model still appears (in line 54), including the text description of the contingency model in the "Discussion" section (in line 300-301). The authors are requested to confirm again. In fact, the entire "Literature Review" section seems to be common-sense management concepts, and its help to readers is limited.
- In the "Research Methods" section, the authors focused on MSME (Micro, small, and medium-sized enterprises), but the authors only visited 5 MSMEs. Obviously, these interviewees are insufficient, and the authors did not state the representativeness of the interviewee. Although the authors say "changed the focus of research from MSMEs to small business in developing countries", the interviewees are still too few. Instead, the authors interviewed many public sector personnel, but the interviews with these public sector personnel did not effectively reflect the research content. This shows that the authors have flaws in the selection of interviewees. I raised this question in the second review comments, but the authors did not respond.
- In the “Results” section, the research results of this manuscript are intuitively obvious to readers. The authors believe that the biggest contribution is to revise the contingency model. But I don't understand why the author wants to correct it? How to fix it? What's wrong with the past model? Is the revised applicability more theoretically malleable? These should be the focus of this article. But I did not find it in this manuscript. This question was also raised in the second review comments, but the authors did not respond.
- In “Discussion” section, this is the most important section of the entire manuscript. However, the authors did not make an in-depth discussion, nor did they make a comparative analysis of past research results. The authors are just restating the results. Such writing does not meet the academic requirements of the "Discussion" section. This question was also raised in the second review comments, but the authors did not respond.
- Figure 1- Figure 5 do not appear in the text of the manuscript, nor are they discussed or explained. This makes readers unable to understand the purpose of these five Figures.
- There are too many spelling and grammatical errors in the authors' manuscripts. There are nearly a hundred of these errors. Even the simplest spelling can make mistakes. This is beyond the scope of the reviewer's assistance. In addition, in the "References" section, the authors have omitted a lot of research in recent years, and there are also many errors in the format. Authors are requested to pay attention to the rigor of writing academic articles. This question was also raised in the second review comments, but the authors did not respond.

Author Response

(The authors gave the same response as above.)

Reviewer 4 Report
While the authors have revised the paper, the discussion section should be expanded.
Author Response

(The authors gave the same response as above.)
